# Switch or Stay? Applying a Push–Pull–Mooring Framework to Evaluate Behavior in E-Grocery Shopping

Terrylina A. Monoarfa [1,*], Ujang Sumarwan [2], Arif I. Suroso [3] and Ririn Wulandari [4]

1   Faculty of Economics, Universitas Negeri Jakarta, Jakarta 13220, Indonesia
2   Department of Family and Consumer Science, IPB University, Bogor 16680, Indonesia
3   School of Business, IPB University, Bogor 16680, Indonesia; arifimamsuroso@apps.ipb.ac.id
4   Faculty of Economics and Business, Universitas Mercu Buana, Jakarta 11650, Indonesia
*   Correspondence: terrylina@unj.ac.id

**Abstract:** This study investigates the antecedents of the intention to switch to e-grocery shopping during the COVID-19 pandemic, which is critical in recommending that e-grocery shopping service providers improve their competitiveness by responding to customer expectations. This study proposes a push–pull–mooring framework to describe the influence of dissatisfaction on the physical market, the attractiveness of e-grocery, and switching costs as factors that drive switching intentions. This study surveyed 252 Indonesians aware of the existence of an e-grocery mobile application, and applied structural equation modeling as an analytical method to explain causal relationships between variables thought to influence switching intentions to e-grocery shopping. The results showed that the attractiveness of e-grocery had a significant effect on switching intention. Likewise, switching costs ultimately drive customer intention to switch to e-grocery shopping. However, dissatisfaction is not a driving factor directly affecting switching costs and intentions. Finally, e-grocery services cannot replace the local tradition of Indonesian people who prefer to shop for groceries in physical markets. Nevertheless, these findings provide theoretical and practical contributions to retail grocery businesses that have integrated conventional and digital services as a future strategy that drives business sustainability.

**Keywords:** switching intention; e-grocery shopping; push–pull–mooring

## 1. Introduction

The i formation search process is the initial stage of the purchase decision [1]. The era of digitalization exposes consumers to various alternative media to access information about goods and services before making a purchase decision. Statista stated that shopping behavior in the Southeast Asia region in 2025 will switch to digital shopping behavior. Online music and video streaming, online learning applications, e-grocery applications, and food delivery services are new products in an emerging digital market. The emergence of COVID-19 seems to have accelerated the process of technology adoption. Based on e-commerce research from the Central Statistics Bureau in 2021, the demand for grocery and food delivery services has increased rapidly during the pandemic. Even compared to other categories, the number of online shopping transactions for groceries and food reaches 30 percent and dominates all online shopping transactions [2]. The pandemic situation and social restriction policies prompted using mobile applications, such as Tanihub, Sayurbox, Cari Sayur, or Happyfresh, to access online grocery shopping. L.E.K. Consulting stated that among several grocery marketing channels, online marketing channels are considered the most strategic channels to penetrate the market [3]. The penetration rate of e-grocery in Indonesia showed an increase in gross merchandise value of USD 1 billion in 2020, so the growth of e-grocery in Indonesia could reach USD 6 billion by 2025. Based on the increasing intention to e-grocery shop and the projection of the development of e-grocery retail business in Indonesia, this study proposes the following research question:

(RQ1) What are the socio-demographic characteristics of potential shoppers who would be interested in switching to e-grocery shopping?

Outside the pandemic situation, Indonesian consumers tend to shop for groceries in traditional markets because of the fresher product quality, relatively affordable prices, and the feeling of bargaining, which is attractive for some consumers [4]. However, some consumers consider the traditional market conditions to be unhygienic. In addition, the traditional market's low food safety encourages customers to switch to modern markets, supermarkets, and even e-grocery [5,6]. In other words, perceived inconvenience is often the reason for dissatisfaction and potentially encourages switching intentions [7,8]. Many studies examined that perceived price fairness, and the availability of delivery services also increases the e-grocery shopping value compared to physical markets [9,10]. Based on the phenomenon that describes the low level of consumer satisfaction with physical market services and the emergence of attractive alternative e-grocery services that offer a variety of shopping conveniences, the following research question is posed: (RQ2) How can physical market dissatisfaction encourage the intention to switch to using e-grocery services, and is the appeal of e-grocery able to encourage consumers to switch from their old habit of physically shopping e-grocery services?

Furthermore, mobile applications' perceived ease of use and usefulness are what makes e-grocery attractive [11]. Finally, the level of personal innovativeness supported the readiness to utilize technological novelty [12].

Customers have their perceptions about the risks of physical shopping, and these might include that physical shopping accelerates the transmission of the virus, which was especially true during the pandemic [13]. In addition, the abundance of health information circulating in various digital media has increased public health consciousness, triggering the avoidance of crowds in the physical market and switching to e-grocery shopping behavior [14]. As a result, at the beginning of the pandemic, the new phenomena of "panic buying" and "pantry stockers" emerged in many households [13]. The demand for online groceries increased rapidly, encouraged scarcity, and pushed the price level. The characteristics of the groceries that most consumers buy include staple food, instant canned food, processed frozen food, and fresh food, such as beef, chicken, fish, seafood, eggs, dairy products, vegetables, and fruit [15]. Then, to prove the importance of health consciousness and personal innovativeness, this study proposes the following question: (RQ3) How important is the role of health consciousness and personal innovativeness as the moderators in influencing switching intention to e-grocery shopping?

Regarding e-grocery shopping consumer behavior and the potential growth of e-grocery businesses in Indonesia, this study aims to: (1) Identify the characteristics of potential customers of e-grocery shopping services; (2) Analyze the effect of perceived dissatisfaction with the physical market and the attractiveness of e-grocery services on switching intentions toward e-grocery shopping; and (3) Analyze the role of moderation of switching costs, health consciousness, and personal innovativeness in accelerating or inhibiting switching intentions toward e-grocery shopping.

The grocery shopping channel preferences are an exciting topic to explore. Some consumers prefer to shop conventionally in a physical market; however, other consumers with modern lifestyles generally prefer the online market in order to have a practical shopping process. Furthermore, will switching behavior to e-grocery shopping only occur during the pandemic, or will it continue post-pandemic? This study theoretically contributes to identifying e-grocery customers' characteristics and analyzing what factors significantly affect switching intention toward online grocery shopping through the push–pull–mooring framework. In addition, this research also contributes practically to recommending the right strategy for retail grocery businesses to maintain business sustainability in the long term.

## 2. Literature Review and Hypothesis

### 2.1. Push–Pull–Mooring Framework

The push–pull–mooring (PPM) framework begins with the law of migration, which explains why a person migrates from the point of origin to the point of destination [16]. Initially, migration studies focused on the concept of aggregate-level push and pull. Push factors are unfavorable conditions that encourage individuals to leave the point of origin, while pull factors are favorable conditions that attract individuals toward the destination point. Then, the researchers considered the possibility of normative and psychosocial variables contributing to migration decisions. So, Moon explained that migration adds to the idea of "mooring", which is then incorporated into the push–pull migration model [17]. The mooring factors are integrated into the push–pull framework to explain migration behavior more thoroughly, where mooring factors can facilitate and inhibit migration behavior. Mooring extends the notion of intervention variables and includes all personal, social, and cultural variables to moderate the migration decision.

Various studies have widely adopted the PPM framework in consumer behavior to analyze the factors that drive consumers to switch to other products (brands) or services. For example, research analyzing switching intention to adopt the food traceability technology has applied the push–pull–mooring framework and integrated it with the DeLone and McLean ISS and TPB Model [18]. The research has found that the integrated model contributes to the relationship quality, encouraging switching intention. Other studies in a similar field highlight willingness to pay a premium price and health consciousness as mooring factors that accelerated switching behavior during the COVID-19 pandemic [14]. This study has found that health consciousness moderates the causality between perceived product quality and switching intention.

Meanwhile, willingness to pay premium price has moderated the causality between perceived risk uncertainty and switching intention, contrary to the research of R. Singh and Rosengren, which analyzed switching intention to online grocery shopping, which focused more on switching costs as a mooring factor in accelerating switching intention [19]. Similarly, research in other fields has highlighted the critical role of switching costs and personal innovativeness in accelerating switching intention [20–24].

### 2.2. Perceived Dissatisfaction with Physical Market as a Push Factor

Dissatisfaction is a driving factor that affects switching intention. Some things that have the potential to cause dissatisfaction are service quality and perceptions of unfair prices [7]. Specifically, in terms of service quality, customers often complain about the performance of customer service, delivery service, and technical service facilities. The customers' complaints reflect the perceived dissatisfaction, thus encouraging customers to switch to another grocery store in the future [19,25]. The higher the attractiveness of competitors' services, the more likely the intention to switch. In the context of electronic services, the quality of information systems is an essential factor that customers consider when choosing electronic payment services [20,21]. In addition, customer assessment of perceived risk is also why customers choose food delivery services [18]. They consider how much they can afford the consequences of the uncertain risk of food delivery service. Another critical reason customers are loyal is the perception of shopping convenience [22]. Convenience is the customer's perception of the time and effort required to use the service [26]. Convenience reflects how service can efficiently complete customers' tasks [27]. Within online retail shopping, customers need convenience reflecting access to applications and secure electronic transactions [28,29].

Furthermore, Clulow and Reimers [30] stated that the condition of the service location (space) is also a factor that determines convenience in the retail business scope. As Maruyama [6] stated, customers switch from traditional markets to modern markets due to the inconvenience of the unhygienic traditional market environment and product quality that is still below standard. Based on previous research, the hypotheses proposed in this study are as follows.

**Hypothesis 1 (H1).** *Perceived dissatisfaction with physical market influences switching cost toward e-grocery shopping.*

**Hypothesis 2 (H2).** *Perceived dissatisfaction with physical market influences switching intention toward e-grocery shopping.*

*2.3. Alternative Attractiveness of E-Grocery as a Pull Factor*

Alternative attractiveness is a positive perception that describes a prospective thing in the new destination, which attracts customers to switch. According to the push–pull–mooring perspective, naturally and unconsciously, when a customer's evaluation results show interest in a new alternative, the switching intention has been raised [16]. Customers switch when the alternative product (brand) or service offers more advantages and benefits than the previous product (brand) or service provider. In other words, the alternative attractiveness becomes a factor that drives switching intention when the provided services are more satisfactory [31,32]. From the perspective of e-services, service quality of information systems will lead to more vital satisfaction that urges the customer to switch to a new service [33]. Lin, Jin, Zhao, Yu, and Su [23] indicated that the ease of use of online learning applications and perceived usefulness are aspects of the pull factors that encourage switching intention to use online learning media. Otherwise, the attractiveness of alternatives contributes to the switching shopping behavior from shopping in physical markets to e-grocery mobile application services [8,26,34,35]. The hypotheses proposed in this study are as follows.

**Hypothesis 3 (H3).** *The alternative attractiveness influences switching cost toward e-grocery shopping.*

**Hypothesis 4 (H4).** *The alternative attractiveness influences switching intention toward e-grocery shopping.*

2.3.1. Perceived Ease of Use and Usefulness

The technology acceptance model predicts whether a new information system can be accepted by potential users [36]. The technology acceptance model mentioned several indicators of the quality of information systems, namely perceived ease of use and usefulness and other supporting external factors. Especially in food delivery applications, perceived ease of use criteria is the ease of learning, controlling, understanding, and being skilled in using the mobile application to order, transact, and pay digitally [35]. Perceived ease of use of an information system indirectly increases the readiness of users to adapt to new technology [36]. For example, during a pandemic, when the e-grocery mobile application is easily accessible, has straightforward navigation, and stands easy to operate, it will encourage consumers to switch to e-grocery services quickly [37]. In addition to perceived ease of use, a mobile application should also provide optimal benefits for users. Several criteria of perceived usefulness include accelerating work completion, improving performance, increasing productivity, increasing work effectiveness, and simplifying work [24,34]. In addition, some customers feel practically supported when using grocery and food delivery services [38,39], as they should not go to the physical market and encounter traffic and crowdedness. The influence of perceived ease of use and usefulness was examined by testing the following hypotheses.

**Hypothesis 5 (H5).** *Perceived ease of use influences e-grocery shopping attractiveness.*

**Hypothesis 6 (H6).** *Perceived usefulness influences e-grocery shopping attractiveness.*

### 2.3.2. Perceived Value

In addition, the attractiveness of e-grocery mobile shopping is not only about the quality of the information system; an e-grocery shopping service should offer a better perceived value than shopping in the physical market. Perceived value represents a trade-off between the quality (benefits) received and the sacrifices (financial, time, and effort) that consumers spend to consume goods and services [40,41]. Specifically, Refs. [32,42] stated that e-service quality is the essential factor impacting high perceived value for online shopping. In the context of retail shopping, the perceived value reflects the product performance and consumer appreciation of the reliability of the service, then compared with the price that must be paid [32].

Meanwhile, other studies stated that product quality, perceived price fairness, and healthy consumption contribute to the perceived value and repurchase intention for shopping organic products [43]. Likewise, Ref. [9] mentioned that perceived price fairness potently encourages perceived value and repurchase intention for e-grocery shopping. Based on previous research, perceived value is an essential factor impacting purchasing decisions on whether to stay (loyal) or prefer other service alternatives (switching). So, the hypothesis proposed in this study is as follows.

**Hypothesis 7 (H7).** *Perceived value influences e-grocery shopping attractiveness.*

### 2.4. Switching Cost toward E-Grocery Shopping as a Mooring Factors

The simplicity of the push–pull perspective turned out to be insufficient to explain the complexity of the factors that cause switching behavior. The concept of contemporary migration explains that evaluating push and pull factors cannot confirm the reasons for switching behavior [44]. Instead, the mooring factor analyzed the possibility of intervention as a trigger to accelerate or inhibit the impulse to switch. One of the mooring factors widely discussed in previous studies is switching costs. Switching cost is an anchoring factor in switching intention. The perceived sacrifice, both financial and non-financial, is often a consideration for someone to accept new behavior. The switching costs include time, effort, sacrifice, the consequences of uncertainty, the amount of coverage for risks, and the ability to adapt to new behavior [45]. When the evaluation results show relatively high switching costs, customers tend to stay loyal to existing ones.

On the contrary, if a new product (brand) and service offer many benefits, the customer can deal with the consequences of switching. Then, the customer's tendency will be quickly switched to a new one [31]. This study summarizes this theoretical and previous research review with the following hypothesis.

**Hypothesis 8 (H8).** *Switching cost influences switching intention toward e-grocery shopping.*

### 2.4.1. Health Consciousness

Health consciousness is the awareness of and concern about self-healthiness, family healthiness, and people's healthiness [46]. Health consciousness assesses an individual's readiness to make various health-related decisions. Health consciousness affects a person's preferences and buying behavior in food products [47,48]. For example, health-conscious individuals consume organic food to improve their quality of life [49–51]. Thus, health-conscious people will be more motivated to turn to credible alternatives. In addition, when consumers are health-oriented, they are more likely to care about product quality and tend to avoid the risk associated with potential health issues [9,14,52]. Based on their knowledge and health consciousness, one is willing to make behavioral changes, including changing grocery shopping habits from physical markets to online grocery shopping [13,48]. During the COVID-19 pandemic, health consciousness is considered one of the factors that minimize switching costs [14]. In addition, people tend to pay more attention to food hygiene and safety, the cleanliness of shopping locations, and service facilities that can

minimize the virus' risk [10]. The influence of health consciousness was examined by testing the following hypothesis.

**Hypothesis 9 (H9).** *Health consciousness influences switching cost toward e-grocery shopping.*

2.4.2. Personal Innovativeness

Personal innovativeness is accepting new ideas in developing information systems technology [53]. Individual beliefs and attitudes toward technologies are critical in predicting behavioral intention [54]. In addition, personal voluntariness and experience also have a role in the intention to adopt technology [55]. Likewise, innovative personal traits and situational encouragement often trigger the transition of someone with technology constraints to become a technology adopter [48,56]. The personal efficacy of technology improves users' innovation in their expertness in IT-related activities [48]. The magnitude of the acceptance ability will encourage the willingness to try new things or the latest digital applications [57]. For example, a study of digital payment stated that personal innovativeness determines the intention to switch to electronic payment services at physical stores [12,19].

In applying the unified theory of acceptance and use of technology (UTAUT) concept, personal innovativeness is a potential factor influencing behavioral intention [58]. During the COVID-19 pandemic, social distancing had become a situational impetus for one to adopt online shopping services. The urge is becoming stronger because personal innovation toward high technology strengthens people's trust in online shopping services [59]. However, limited studies still explain that personal innovativeness can minimize switching costs. Therefore, the higher the personal innovativeness of technological novelty, the smaller the perceived sacrifice to make changes (switching cost). Based on theoretical studies and previous research, the proposed hypothesis is as follows.

**Hypothesis 10 (H10).** *Personal innovativeness influences switching cost toward e-grocery shopping.*

2.4.3. The Role of Mooring Factor as a Moderator

Based on a push–pull–mooring perspective, the mooring factor does not directly affect switching intention. However, the variables included in the mooring factor often take on the role of moderators that facilitate switching behavior. For example, a study at the beginning of the pandemic stated that health consciousness was essential in determining the location and media used to shop for groceries [13]. Another study stated that health consciousness significantly encourages the emergence of purchase intentions for organic products [48]. However, this study has not researched the possible role of health consciousness as a moderating variable. Furthermore, research on traceable agricultural products has placed health consciousness as a dimension of the mooring factor that moderates the causality of push–pull factors on switching intention [14]. Based on a review of previous research, this study proposes a hypothesis about the moderating effect of health consciousness on the causality of switching costs on switching intention towards e-grocery shopping.

Handarkho and Harjoseputro [12] have investigated the role of personal innovation as a mooring factor in driving the intensity of the use of electronic payment services in a physical store, which has a significant effect. However, this study has not tested the moderating effect of personal innovativeness on the intention of using electronic payment services. However, other studies on payment systems have seen the potential for personal innovation as part of the mooring factor that simultaneously moderates the causality of the driving factors for the intention to switch [19]. Therefore, based on previous research, this study proposes a hypothesis to test personal innovation's ability to moderate the push–pull factors on switching intentions to use e-grocery shopping services.

Another factor suspected of being a mooring factor is switching costs. Various previous studies have investigated the role of switching cost in determining switching intention [19,31,48]. Moreover, consumers will consider switching behavior's consequences, constraints, and

risks. Furthermore, this study will explore how moderate switching costs affect the causality of push–pull factors on the switching intention. According to the previous research review, the proposed hypotheses are as follows.

**Hypothesis 11 (H11).** *Switching costs moderate the causal relationship of perceived dissatisfaction with physical markets to switching intention toward e-grocery shopping.*

**Hypothesis 12 (H12).** *Switching costs moderate the causal relationship of alternative attractiveness to switching intention toward e-grocery shopping.*

**Hypothesis 13 (H13).** *Health consciousness moderates the causal relationship of switching cost to switching intention toward e-grocery shopping.*

**Hypothesis 14 (H14).** *Personal innovativeness moderates the causal relationship of switching cost to switching intention toward e-grocery shopping.*

## 3. Methodology

### 3.1. Sampling and Data Collection

This quantitative research used the survey method and the non-probability sampling approach in which each individual has an unequal opportunity to be a selected respondent (sample). The sample selection process uses the purposive sampling technique, a combination of convenience and judgmental sampling based on considerations of ease of collecting sampling. Furthermore, the researcher has set specific criteria for recruiting respondents [60]. This study distributed questionnaires online through social media to obtain the respondents who matched the criteria. The researcher set sample criteria in the screening question section of the questionnaire that the respondents are residents of Jakarta, Indonesia, who were aware of the existence of e-grocery shopping services through mobile applications, such as Tanihub, Sayurbox, Cari Sayur, or Happyfresh. The sample size for SEM analysis is a minimum of 200 samples, or 5–10 times the number of indicators [61,62]. The number of instruments (items) used in this study was 39. Through the distribution of online questionnaires, 265 Jakarta residents filled out the questionnaire. However, the result of the final sample was 252 respondents for empirical analysis. So, it can be concluded that the number of samples meets the sample size criteria.

According to the data collection, the sample's socio-demographic characteristics comprised 193 (76.6 percent) women and 59 (23.4 percent) men. Regarding the age group, most of the respondents in the sample are aged 25–30 years and between 31–40 years, and most respondents are married. In addition, most of the respondents' education level is undergraduate, and they work as employees with an average monthly income of between 400 and USD 700. The socio-demographic characteristics of respondents are relevant to the Ken Research of 4 April 2022, which stated the demand for online grocery shopping mainly comes from working professionals and married couples because they have less time to visit stores to buy groceries physically. Additionally, most are housewives or wives who care for the family's food needs. About 49.7 percent of the total population in Indonesia are women, so it makes sense that most e-grocery customers are women. Furthermore, the customers who make about 70 percent of grocery orders are 25–37 years old [63]. All the information regarding the socio-demographic characteristics of the respondents is presented in Table 1.

**Table 1.** Socio-demographic characteristics of respondents.

| Socio-Demographic Items | Frequency | Percentage |
|---|---|---|
| | **n = 252** | |
| Gender | | |
| Male | 59 | 23.4 |
| Female | 193 | 76.6 |

**Table 1.** *Cont.*

| Socio-Demographic Items | Frequency | Percentage |
|---|---|---|
| | **n = 252** | |
| Marital Status | | |
| Single | 87 | 34.5 |
| Married | 164 | 65.1 |
| Divorced/Widowed | 1 | 0.4 |
| Age | | |
| 25–30 years old | 181 | 71.8 |
| 30–39 years old | 56 | 22.2 |
| 40–49 years old | 11 | 4.4 |
| 50 years old and over | 4 | 1.6 |
| Education | | |
| High school degree | 20 | 8.0 |
| Under-graduate | 194 | 77.0 |
| Post-graduate | 38 | 15.0 |
| Occupation | | |
| Self-employed | 69 | 27.4 |
| Employee | 126 | 50.0 |
| Housewife | 57 | 22.6 |
| Monthly Income | | |
| Under USD 400 | 27 | 10.7 |
| Between USD 400–700 | 128 | 50.8 |
| Over USD 700 | 97 | 38.5 |

*3.2. Research Instrument*

The questionnaire was organized into two sections. The first section included six items to classify the characteristic of the socio-demographic characteristics of the respondents, such as gender, marital status, age, education level, occupation, and average monthly income. The second section measures consumer switching behavior, including 39 items on the construct, which were modified and adapted from a previous study.

For example, perceived dissatisfaction with the physical market as a push factor has eight items adapted from [10,24,31]; alternative attractiveness as a pull factor has four items adapted from [8]; perceived ease of use has four items adapted from [24,34]; perceived usefulness has four items adapted from [24,34]; perceived value has four items adapted from [9,10,33]; switching costs as a mooring factor has four items adapted from [19]; health consciousness has five items adapted from [47]; personal innovativeness has four items adapted from [19]; and switching intentions has three items adapted from [19]. The research questionnaire is attached in Appendix A.

The questionnaire proposed several items that describe the measurement of variables and asked respondents to provide an assessment based on a Likert scale, which describes the degree of agreement. It starts from "Strongly Disagree" and ends with "Strongly Agree", which are worth 1 to 5 points. Furthermore, the respondents' assessment will be analyzed to measure the ability of the independent variable to explain the dependent variable. Previously, this study conducted a pilot study to test the validity and reliability of the instrument on 30 respondents. All the proposed hypotheses in the conceptual model are depicted in Figure 1.

*3.3. Analytical Method*

This study used covariance-based structural equation modeling (CB-SEM) as the analytical method capable of simultaneously estimating several separate but interconnected regressions [62]. This study conducted several tests and analyses using the statistical package for the social science (SPSS) and the analysis of moment structure (AMOS) software. There is a three-step approach to conducting the data analysis. First, the validity and

reliability construct was assessed by conducting confirmatory factor analysis (CFA). Second, the goodness-of-fit testing ensures that the structural model meets the criteria of absolute indices. Third, the proposed hypotheses were conducted by structural equation modeling (SEM) to examine the structural model and moderating effects.

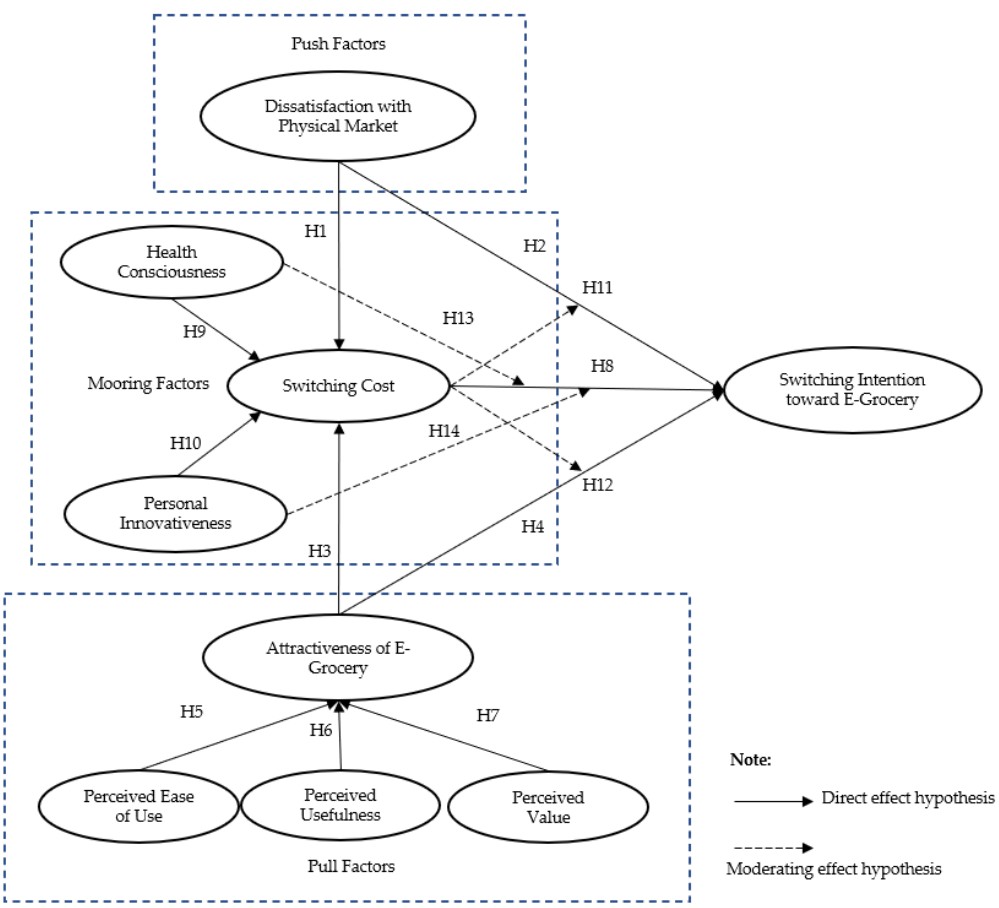

**Figure 1.** The Conceptual Framework.

## 4. Data Analysis and Result

In this study, confirmatory factor analysis (CFA) examined the measurement model fit to test the validity of the convergent and discriminant latent factors. The results of the CFA declared that all items were valid with a loading factor value above 0.5 and reliable with a Cronbach's alpha value above 0.6. As its next step, this research should ensure that the structural model has goodness of fit by considering the value of the absolute indices component based on certain criteria. Several absolute index components, namely chi-square ($\chi 2$) with a probability cut-off value of >0.05; root mean square error of approximation (RMSEA) with a cut-off value of <0.08; goodness of fit index (GFI) with a cut-off value of 0.90; adjusted goodness of fit (AGFI) with a limit value of 0.90; CMIN/DF with a limit value < 2.0; the Tucker–Lewis index (TLI) with a cut-off value > 0.95, and a comparative fit index (CFI) with a cut-off value of >0.95. This study eliminated the items with a high standard error in the modification indices to ensure the structural model achieves goodness of fit. At the same time, five items were eliminated: one item on perceived dissatisfaction, one item on perceived usefulness, one item on switching cost, and two items on personal innovativeness. By eliminating items on modification indices, the structural model achieved a good fit in which the value of the chi-square was 505.826 with the probability value being 0.174; RMSEA was 0.016; GFI was 0.901; AGFI was 0.876; CMIN/DF was 1.060; TLI was 0.996; CFI was 0.997. The result of the CFA after eliminating the items is reported in Table 2.

**Table 2.** Confirmatory factor analysis.

| Variables | Items | Loading Factor | Cronbach's Alpha |
|---|---|---|---|
| Perceived dissatisfaction with physical market | DSAT1 | 0.952 | 0.936 |
| | DSAT2 | 0.937 | |
| | DSAT3 | 0.939 | |
| | DSAT4 | 0.833 | |
| | DSAT5 | 0.885 | |
| | DSAT6 | 0.887 | |
| | DSAT8 | 0.818 | |
| Alternative attractiveness | ATT1 | 0.925 | 0.939 |
| | ATT2 | 0.891 | |
| | ATT3 | 0.926 | |
| | ATT4 | 0.935 | |
| Perceived ease of use | EAS1 | 0.948 | 0.962 |
| | EAS2 | 0.957 | |
| | EAS3 | 0.943 | |
| | EAS4 | 0.948 | |
| Perceived usefulness | USE1 | 0.933 | 0.955 |
| | USE2 | 0.941 | |
| | USE3 | 0.942 | |
| Perceived value | VAL1 | 0.903 | 0.918 |
| | VAL2 | 0.874 | |
| | VAL3 | 0.904 | |
| | VAL4 | 0.904 | |
| Switching cost toward e-grocery shopping | SWC1 | 0.903 | 0.929 |
| | SWC2 | 0.898 | |
| | SWC3 | 0.912 | |
| Health consciousness | HEA1 | 0.805 | 0.812 |
| | HEA2 | 0.826 | |
| | HEA3 | 0.798 | |
| | HEA4 | 0.767 | |
| Personal innovativeness | INO2 | 0.908 | 0.911 |
| | INO3 | 0.916 | |
| Switching intention toward e-grocery shopping | SWI1 | 0.896 | 0.901 |
| | SWI2 | 0.915 | |
| | SWI3 | 0.932 | |

Furthermore, this study conducted hypothesis testing based on the regression estimation results. The causality relationship among variables is declared significant if the probability value is below 0.05 and the C.R. value above 1.96. As a result, there are two hypotheses proposed that were rejected. As a result, H1, which proposed that the causal relationship of perceived dissatisfaction with the physical market to switching costs toward e-grocery shopping, has a probability value of 0.263 (above 0.05) and a C.R. value of 1.120 (below 1.96), so it can be concluded that H1 is rejected. Then, H2, which proposed the causal relationship of perceived dissatisfaction with the physical market to switching intention toward e-grocery shopping, has a probability value of 0.069 (above 0.05) and a C.R. value of 1.817 (below 1.96), so it can be concluded that H2 is rejected. However, other hypotheses that were proposed are supported. The result of the structural model is depicted in Figure 2.

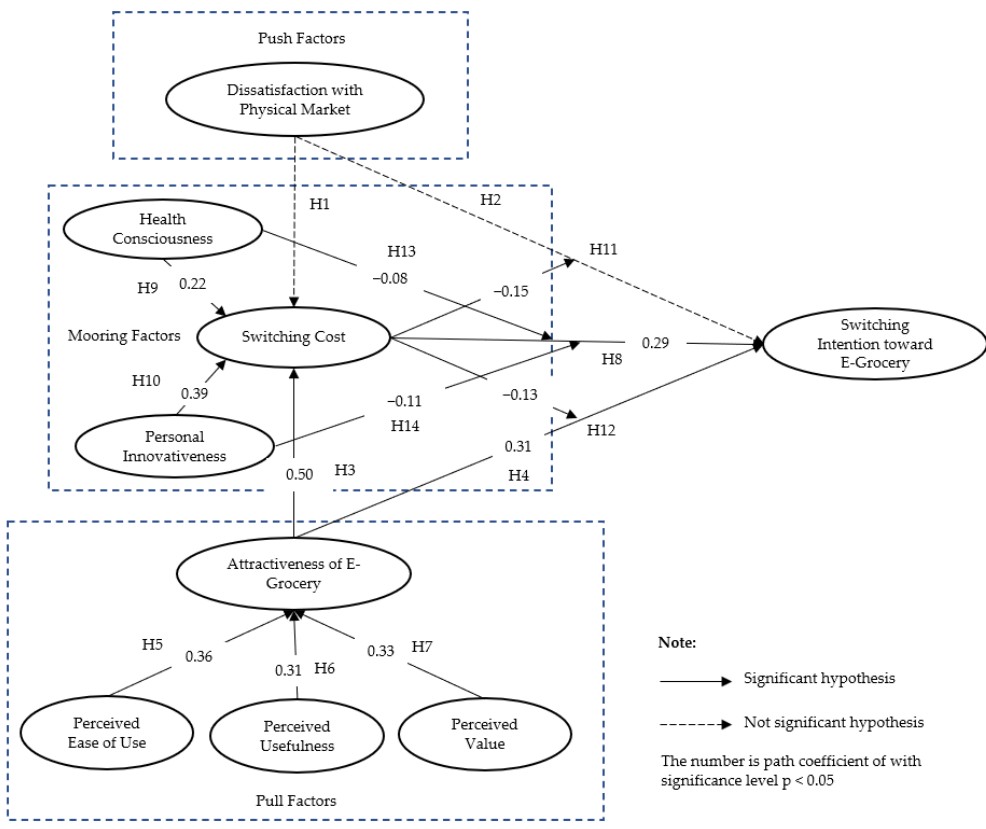

**Figure 2.** The Result of Structural Model.

The result of the proposed hypotheses and regression estimation are reported in Table 3.

**Table 3.** Estimation regression of structural model analysis.

| | Hypotheses | | | Estimate | S.E. | C.R. | *p* | Result |
|---|---|---|---|---|---|---|---|---|
| H1. | DSAT | → | SWC | 0.087 | 0.077 | 1.120 | 0.263 | Not supported |
| H2. | DSAT | → | SWI | 0.127 | 0.070 | 1.817 | 0.069 | Not supported |
| H3. | ATT | → | SWC | 0.504 | 0.111 | 4.550 | *** | Supported |
| H4. | ATT | → | SWI | 0.314 | 0.129 | 2.435 | 0.015 | Supported |
| H5. | EAS | → | ATT | 0.358 | 0.123 | 2.919 | 0.004 | Supported |
| H6. | USE | → | ATT | 0.314 | 0.110 | 2.855 | 0.004 | Supported |
| H7. | VAL | → | ATT | 0.332 | 0.062 | 5.382 | *** | Supported |
| H8. | SWC | → | SWI | 0.294 | 0.123 | 2.388 | 0.017 | Supported |
| H9. | HEA | → | SWC | 0.220 | 0.104 | 2.116 | 0.034 | Supported |
| H10. | INO | → | SWC | 0.394 | 0.095 | 4.152 | *** | Supported |

Notes: DSAT = dissatisfaction, ATT = alternative attractiveness, EAS = perceived ease of use, USE = perceived usefulness, VAL = perceived value, SWC = switching cost, HEA = health consciousness, INO = personal innovativeness, and SWI = switching intention. Significance level *p* < 0.05 and *** for significance level *p* < 0.001.

In addition to the hypothesis testing of the direct effect, this study also conducted hypothesis testing for moderating effect. To examine the hypothesis proposed regarding the moderating effect, previously, we had to calculate the multiplication of the z-score of the moderator variable with the z-score of the independent variable that affected the dependent variable. Then, the moderation model was calculated on the AMOS software to obtain the regression estimation results. As a result, the proposed hypotheses of the moderation model are supported. The result of the hypotheses of the moderation effect is shown in Table 4.

**Table 4.** Estimation regression of moderation structure.

| | Hypothesis of Moderating Effect | | | Estimate | S.E. | C.R. | *p* | Result |
|---|---|---|---|---|---|---|---|---|
| H11. | SWC × DSAT | → | SWI | −0.151 | 0.034 | −4.407 | *** | Reversely Supported |
| H12. | SWC × ATT | → | SWI | −0.129 | 0.036 | −3.604 | *** | Reversely Supported |
| H13. | HEA × SWC | → | SWI | −0.076 | 0.030 | −2.576 | 0.010 | Reversely Supported |
| H14. | INO × SWC | → | SWI | −0.112 | 0.040 | −2.780 | 0.005 | Reversely Supported |

Notes: DSAT = perceived dissatisfaction, ATT = alternative attractiveness, HEA = health consciousness, INO = personal innovativeness, SWC = switching cost, and SWI = switching intention. Significance level $p < 0.05$ and *** for significance level $p < 0.001$.

## 5. Discussion and Conclusions

This study developed and examined the factors contributing to the intention to switch to e-grocery shopping through a push–pull–mooring framework. Through the PPM framework, the discussion does not only focus on the low attractiveness of the origin point and the high attractiveness of the destination point (alternative), the more crucial mooring factor is also considered to trigger switching acceleration. Research findings suggest dissatisfaction with the physical market does not motivate switching to e-grocery shopping services. People complain about the inconvenience of traditional markets, but it does not make them want to switch to e-grocery services [19]. Even during the COVID-19 pandemic, there is a high risk of shopping in physical markets, but it does not encourage customers to switch to using e-grocery services.

Thus, the findings reject H1 and H2, which state that perceived dissatisfaction with the physical market does not affect switching costs and intentions toward e-grocery shopping. The pandemic is pushing customers to shop from alternative places and media [64]. However, this does not mean that customers will quickly accept the consequences of switching (switching cost) from the old habit of shopping for groceries in physical markets to e-grocery services [45]. Especially in Indonesia, e-grocery services seem to be an alternative shopping service that supports meeting customer needs in certain situations. However, this does not necessarily eliminate consumer interest in conventional shopping, which provides personal transactions between sellers and buyers. The emergence of a shift in shopping behavior through e-grocery is more due to customer volunteerism, particularly for people with high health consciousness and personal innovativeness, so they tend to overlook the switching cost to e-grocery services [12,14,19].

In addition, the result stated that the attractiveness of e-grocery shopping as the pull factor has a binding effect on the switching cost and intention to switch to e-grocery shopping. The finding stated that the attractiveness of e-grocery has the most substantial effect on switching costs compared to other factors, such as health consciousness and personal innovativeness, with an estimated coefficient value of 0.504. Likewise, the finding mentioned that e-grocery attractiveness as a pull factor significantly affects increasing the intention to use e-grocery services. The more attractive the impression of the attractiveness of e-grocery shopping, the easier for customers to accept the switching cost (consequences) and switch to e-grocery services. Among several factors that support the attractiveness of e-grocery services, perceived ease of use is the most decisive factor contributing to the attractiveness of e-grocery services, with a coefficient estimated value of 0.358. It means that e-grocery shopping service providers must focus on increasing the quality of e-grocery applications and ensure that the application is user-friendly. Likewise, e-grocery applications need to optimize customer benefits by developing service features tailored to customer tastes and needs [38]. Furthermore, several factors to improve the perceived value of e-grocery shopping are product quality, availability, delivery service, and perceived price fairness [9,10]. Even others that cannot be measured financially, such as the company's reputation, brand image, or company credibility, also play a role as contributors to high perceived value [31,41].

Regarding the role of the mooring factor, the finding mentioned that switching costs contributed to the switching intention toward e-grocery shopping. Switching costs consist

of monetary or non-monetary costs incurred due to switching. Switching cost has a role that is sensitive to switching intention. Every switching behavior will have its consequences. Nevertheless, the magnitude of the consequences depends on how a person responds to a change. In the case of e-grocery shopping, some of the monetary switching costs incurred could be product prices and shipping costs. At the same time, non-monetary switching costs could be the time to adapt to learn the use of electronic ordering systems and the risk of uncertainty in delivery times [45]. However, for people with high personal innovativeness, these non-monetary transfer costs can be easily overcome [55]. Likewise, people with a high level of health consciousness tend to have personal triggers that encourage them to ignore switching costs, so they immediately switch to e-grocery services [64].

Switching shopping behavior often impacts customers during the adaptation process [8]. Switching towards e-grocery shopping requires consumers to understand how to use e-grocery mobile applications, how the shopping procedures work, and various possible uncertainties that cannot be predicted in advance and arise as part of the consequences of e-transactions [10,12,19]. Health consciousness and personal innovativeness significantly affect switching costs toward e-grocery shopping. Interestingly, the finding showed that personal innovativeness plays a vital role in minimizing switching costs compared to health consciousness, with an estimated coefficient value of 0.394. Individuals with high curiosity and innovativeness towards technological novelty make it easier to adapt to using e-grocery shopping services [12,19]. However, this study has not yet tested the personal innovativeness encouraging switching intentions. In addition to personal innovativeness, the finding showed that health consciousness also contributed to the acceptance of switching costs. The pandemic provided extraordinary knowledge and experience about the importance of health [13]. At that time, health consciousness became a motivation for engaging with the switching costs and minimizing the switching costs that arise [13,14,48], then encouraging customers to switch to e-grocery services.

The mooring factor moderates and can speed up and inhibit switching intentions. The regression of the moderation structure estimation shows that switching costs' contribution to the causality of perceived dissatisfaction with the physical market and switching intention is greater than the attractiveness of e-grocery services, with an estimated coefficient of $-0.151$. Perceived dissatisfaction with physical market services is a reason for one to reduce the value of switching costs, thereby accelerating the intention to switch to e-grocery. Other studies found that switching costs are sensitive in determining the intention to use e-learning and digital payment [19,31]. Therefore, the magnitude of the switching costs has a relative value; perceived dissatisfaction with the physical market will be a strong impetus for switching to e-grocery. The motivation will reduce the cost of switching and overlook the obstacles due to moving. Likewise, the moderating role of switching costs on the causality of the attractiveness of e-grocery services on switching intentions indicates that the higher the perceived attractiveness of e-grocery shopping the lower the risks and possible consequences of switching. The interest in the convenience and benefits of electronic services and an improved value of e-grocery shopping causes one to disregard the consequences of the risks that may arise from online shopping [31,49]. Thus, customers tend to switch to e-grocery shopping services immediately.

This study has positioned health consciousness and personal innovativeness as important moderating variables influencing the causality of switching costs on switching intention. This research has proven that personal innovativeness is a crucial trigger in the causality of switching costs to the switching intention toward e-grocery shopping. The estimated regression value of personal innovativeness as a moderator is $-0.076$, better than health consciousness. Personal innovativeness in adopting technology contributes to minimizing the obstacles of operating an e-grocery application. In addition, it will encourage the adaptation process to be more straightforward in switching to e-grocery shopping.

Furthermore, during the pandemic, health consciousness has become a reason to accept the switching costs. People ignore various barriers to using e-grocery services based

on their knowledge and health consciousness. As a result, customers quickly adapted to various information systems applications to meet their needs continuously.

## 6. Managerial Implication, Limitation, and Future Research Direction

The application of the push–pull–mooring framework shows that the attractiveness of e-grocery as a pull factor plays a vital role in the intention to switch to e-grocery shopping. All e-grocery attractiveness factors, such as perceived ease of use, usability, and shopping value, have a significant effect. Research findings contribute to building modern grocery retail businesses' competitiveness and customer attractiveness through developing e-grocery services to maintain business continuity and long-term customer loyalty. The attractiveness of e-grocery shopping will not only arise due to situational encouragement from the COVID-19 pandemic, but customers will still be interested in using e-grocery shopping services in the post-pandemic period. For this reason, the results provide recommendations to e-grocery shopping providers to improve the system's quality to make it more accessible and provide straightforward navigation. In addition, the interface of the e-grocery application should be more attractive and emphasize improving the shopping experience, which indicates an increase in the perceived value of e-grocery shopping. In addition, developing new features is also one of the recommendations to optimize the specific benefits of e-grocery shopping compared to shopping in the physical market. The Indonesia Online Grocery Market Outlook to 2026 report from Ken Research stated that e-grocery service providers need to prepare a strategy to respond to the growth of the e-grocery market in Indonesia, population growth, and high internet penetration for online shopping [65].

Based on the socio-demographic characteristics of the respondents, most of the e-grocery shopping enthusiasts are millennial consumers with tech-savvy characteristics. They tend to have highly innovative personalities and quickly adapt to new technologies. In addition, e-grocery services also need to highlight the potential for customer health awareness to pay more attention to product freshness, food safety standards, and hygiene as a market attraction and unique selling points of e-grocery services. Likewise, the timeliness of delivery services and the flexibility of delivery schedules are the main attractions of e-grocery services for urban communities compared to the value of shopping in the physical market.

Despite the various findings and managerial implications presented, this study still has limitations in certain aspects. First, the findings represent consumer spending behavior in Indonesia, so there is still potential for different findings if further research applies the conceptual model in other countries or even in different research fields. However, other studies can utilize the result as a reference for analyzing e-grocery shopping behavior on the characteristics of society and culture that are almost similar to Indonesia, such as countries in Southeast Asia where the e-grocery retail industry is an emerging market. Second, this study is limited to examining certain factors that influence switching intentions towards e-grocery shopping in Indonesia during the pandemic, as the period of consumer surveys present in a conceptual research model. For this reason, further research could include other variables driving the intention to switch to e-grocery shopping. Third, this study has not tested the role of respondents' socio-demographic characteristics in shopping behavior and preferences for e-grocery services. Thus, further research can comprehensively analyze and formulate groceries retail businesses' marketing strategies.

**Author Contributions:** Writing–original draft, T.A.M.; Writing–review & editing, T.A.M.; Supervision, U.S., A.I.S. and R.W. All authors have read and agreed to the published version of the manuscript.

**Funding:** This research received no external funding.

**Institutional Review Board Statement:** Not applicable.

**Informed Consent Statement:** Not applicable.

**Data Availability Statement:** The data presented in this study are available on request from the corresponding author.

**Conflicts of Interest:** The authors declare no conflict of interest.

## Appendix A

**Table A1.** Research Questionnaire.

| Variables and Items | References |
|---|---|
| **Push Factors—Perceived Dissatisfaction (DSAT)**<br>DSAT1. I feel unhappy making purchases in the physical market.<br>DSAT2. The crowd of physical markets makes me uncomfortable.<br>DSAT3. I am worried about the cleanliness of the physical market during the COVID-19 pandemic.<br>DSAT4. Shopping at the physical market wastes my time.<br>DSAT5. I cannot flexibly arrange my schedule to shop in the physical market.<br>DSAT6. I can arrive at the physical market without effort (R).<br>DSAT7. The environment of the physical market is appropriate to my situation during COVID-19 pandemic.<br>DSAT8. Overall, I feel dissatisfied shopping in the physical market. | [10,24,31] |
| **Pull Factors—Alternative Attractiveness (ATT)**<br>ATT1. E-grocery shopping has better offers than physical market.<br>ATT2. The service performance on e-grocery shopping is more interesting than physical market.<br>ATT3. The e-grocery shopping application is effective to meet my needs.<br>ATT4. Overall, the e-grocery shopping application is more exciting than physical market. | [8] |
| **Perceived Usefulness (USE)**<br>USE1. Using e-grocery shopping application helps me accomplish thing so quickly.<br>USE2. Using e-grocery shopping application increases my productivity.<br>USE3. Using e-grocery shopping application enhances my effectiveness.<br>USE4. Overall, using e-grocery shopping application give me a benefit. | [24,34] |
| **Perceived Ease of Use (EAS)**<br>EAS1. Learning to use the e-grocery shopping application is easy for me.<br>EAS2. There is a clear and understandable navigation at the e-grocery shopping application.<br>EAS3. It is easy for me to become skillful at using the e-grocery shopping application.<br>EAS4. Overall, the e-grocery shopping application is easy to use. | [24,34] |
| **Perceived Value (VAL)**<br>VAL1. Compared to physical market, the product's price at the e-grocery shopping application is acceptable.<br>VAL2. Compared to physical market, the product's price at the e-grocery shopping application is very economical.<br>VAL3. Compared to physical market, the product at e-the grocery shopping application has a good value.<br>VAL4. The e-grocery shopping application has a good level of service performance for the money I spend. | [9,10,33] |
| **Mooring Factors—Switching cost (SWC)**<br>SWC1. It would take a lot of time changing to e-grocery shopping.<br>SWC2. It would take a lot of effort changing to e-grocery shopping.<br>SWC3. It would take a lot of learning costs to switch to e-grocery shopping.<br>SWC4. In general, it would be a hassle changing to e-grocery shopping. | [19] |

**Table A1.** *Cont.*

| Variables and Items | References |
| --- | --- |
| **Health Consciousness (HEA)**<br>HEA1. I am very conscious about my health and the health of others for whom I shop.<br>HEA2. I assume accountability for the state of my health and others in the household for whom I shop.<br>HEA3. I am very involved with my health and the health of others for whom I shop.<br>HEA4. I am very concerned about the number of artificial preservatives in food.<br>HEA5. The safety of food nowadays concerns me a lot. | [47] |
| **Personal Innovativeness**<br>INO1. If I heard about a new information technology, I would look for ways to experiment with it.<br>INO2. Among my peers, I am usually the first to try out new information technologies.<br>INO3. For me, experimenting with new technologies is challenging.<br>INO4. I like to experiment with new information technologies. | [19] |
| **Switching intention (SWI)**<br>SWI1: I plan to use e-grocery shopping in the future.<br>SWI2: I will reduce my physical grocery shopping to e-grocery shopping.<br>SWI3: E-grocery shopping is likely to become the primary shopping method of mine in the future. | [19] |

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
