# Peer review of "Switch or Stay? Applying a Push–Pull–Mooring Framework to Evaluate Behavior in E-Grocery Shopping"

_sustainability, doi:10.3390/su15076018_

Round 1

Reviewer 1 Report (Previous Reviewer 3)

I am happy with the revised manuscript.

Author Response

Dear Reviewer, 

Thank you for your prompt review of the manuscript. I am grateful to the reviewers for the constructive comments. I appreciate the opportunity to improve my manuscript and found the comments and suggestions very helpful in revising the manuscript. In response to your E-mail from March 8, 2023, concerning the manuscript ID: sustainability-2224385, entitled "Switch or stay? Applying a push-pull-mooring framework to evaluate behavior in e-grocery shopping", I have revised the manuscript to address all the concerns raised by reviewers. The following are the changes made in the revised manuscript:

To address the English language's quality, I have corrected my English in many parts of the manuscript. I also added a clear statement of research questions in the introduction section, which aligned with the research objectives. Then, the suggestion to explain the research design and how the researcher conducts the online survey are clearly stated in the methodology subsection sampling and data collection. Finally, the point-by-point reviewers’ comments are followed by the author's responses. I believe the abovementioned efforts have significantly improved this revised manuscript and hope it is acceptable for publication in your journal.

Thank you in advance for your efforts on this revised manuscript.

Sincerely, 

Terrylina A. Monoarfa

Reviewer 2 Report (New Reviewer)

The paper is much more better after the mdoifications

Author Response

Dear Editor, 

Thank you for your prompt review of the manuscript. I am grateful to the reviewers for the constructive comments. I appreciate the opportunity to improve my manuscript and found the comments and suggestions very helpful in revising the manuscript. In response to your E-mail from March 8, 2023, concerning the manuscript ID: sustainability-2224385, entitled "Switch or stay? Applying a push-pull-mooring framework to evaluate behavior in e-grocery shopping", I have revised the manuscript to address all the concerns raised by reviewers. The following are the changes made in the revised manuscript:

To address the English language's quality, I have corrected my English in many parts of the manuscript. I also added a clear statement of research questions in the introduction section, which aligned with the research objectives. Then, the suggestion to explain the research design and how the researcher conducts the online survey are clearly stated in the methodology subsection sampling and data collection. Finally, the point-by-point reviewers’ comments are followed by the author's responses. I believe the abovementioned efforts have significantly improved this revised manuscript and hope it is acceptable for publication in your journal.

Thank you in advance for your efforts on this revised manuscript.

Sincerely, 

Terrylina A. Monoarfa

Reviewer 3 Report (New Reviewer)

The paper reply to the reviewers' comments and make correction for improvement.  My recommendation is accepted after minor revision with spell check. Well done.

Author Response

Dear Reviewer, 

Thank you for your prompt review of the manuscript. I am grateful to the reviewers for the constructive comments. I appreciate the opportunity to improve my manuscript and found the comments and suggestions very helpful in revising the manuscript. In response to your E-mail from March 8, 2023, concerning the manuscript ID: sustainability-2224385, entitled "Switch or stay? Applying a push-pull-mooring framework to evaluate behavior in e-grocery shopping", I have revised the manuscript to address all the concerns raised by reviewers. The following are the changes made in the revised manuscript:

To address the English language's quality, I have corrected my English in many parts of the manuscript. I also added a clear statement of research questions in the introduction section, which aligned with the research objectives. Then, the suggestion to explain the research design and how the researcher conducts the online survey are clearly stated in the methodology subsection sampling and data collection. Finally, the point-by-point reviewers’ comments are followed by the author's responses. I believe the abovementioned efforts have significantly improved this revised manuscript and hope it is acceptable for publication in your journal.

Thank you in advance for your efforts on this revised manuscript.

Sincerely, 

Terrylina A. Monoarfa

This manuscript is a resubmission of an earlier submission. The following is a list of the peer review reports and author responses from that submission.

Round 1

Reviewer 1 Report

In my opinion the reviewed article is characterised by a satisfactory level of content. The research gap has been correctly identified and the hypotheses have their references in the literature. The way they are formulated is not objectionable. I have no comments on the applied research method, the description of the obtained results and confronting them with the results of other studies (Discussion section). The authors correctly identified potential addressees of the study results, as well as further directions of analysis and the limitations. Nevertheless, the following points, in my opinion, require further clarification:

1) The survey sample - to what extent is it representative and truly reflective of e-grocery customers (does it cover children? - the first group as regards Age criterion is defined as "under 30 years old", possibly the structure of Indonesian society (if not it requires the strong justification),

2) The study relates to the COVID-19 pandemic period, when, however, consumer behaviour was largely determined by sanitary restrictions. It would be useful to make the case that changes in consumer behaviour are voluntary and not driven by the aforementioned restrictions

3) The concept of switching cost and its quantification needs to be described more broadly. It would also be worth deepening the analysis towards examining the sensitivity of changes in consumer behaviour to a change in switching cost

4) It would also be worth answering the question to what extent the changes in consumer behaviour induced by the COVID-19 pandemic are stable and will not change as soon as the pandemic ends

5) In the context of the COVID-19 experience and the development of e-commerce in a broad sense (including online shopping transactions for groceries and food), it would have been good to provide a synthetic SWOT analysis for e-grocery and physical market shopping

6) The cultural context needs to be clarified in order to be able to answer the question of whether and possibly why the conclusions of the study conducted on the Indonesian market can be generalised, i.e. applied to other societies.  

Reviewer 2 Report

The work presents an adequate conceptual basis. However, there is a lack of associating the conceptual basis with the work's proposal to identify the research gap in the research explicitly. To this end, the authors must create a table identifying the most relevant articles in the area and how the manuscript complements the state-of-the-art with its research proposal.

In section 2.1, lines 91, 92, and 93, it is necessary to present what PPM studies influence consumers in switching their preferences on the most relevant products or services, identifying the bibliographic sources.

In section 3.1, when describing the sample, there is a heterogeneous distribution in the respondent's sociodemographic characteristics considered in the sample, with 76.6% of women. This distribution may represent an undesirable bias in the sample, considering the possibility of transactions made by men. There may also be bias in the research results considering that 71.8% of the sample comprises people under the age of 30 years. This age group tends to use e-commerce more; in this sense, the research's results can be influenced by this bias. The conclusions presented in lines 493 and 494 confirm this limitation.

It is necessary to transform the currency presented in Table 1 into dollars, to facilitate the understanding of the consumer's income included in the sample. 

In section 3.2 (line 320), when it is stated that the research instrument is based on a previous study, it is unclear whether it refers to the study presented in line 332 or to research carried out previously. If the latter is the case, it is necessary to present the bibliographic source.

Reviewer 3 Report

The title of the manuscript seems to be appropriate, and the content of the manuscript is suitable for the journal. The manuscript is about investigating the antecedents of switching intentions toward e-grocery shopping during the COVID-19 pandemic using the push-pull-mooring framework.

A brief sentence about the need for the study should be included at the beginning of the abstract. The introduction section is inadequately described. The objective of the study is not made explicit, and there is no information about the adopted research method. The context of the study is inadequately described, and the purpose of the study needs to be justified clearly.

The literature review section needs to be strengthened by including the latest articles relating to the effect of COVID-19 on online shopping behaviour and e-grocery shopping in general. The gaps in the existing literature should be made clear.

The methodology section gives an inadequate level of detail on the methods employed for the data analysis. For instance, there is no information about how the respondents were recruited to participate in the survey. Also, the provided rationale for the sample size adopted in the study is not adequate. The analysis results have been inadequately explained.

Some of the references are not cited and listed correctly. The manuscript is not written satisfactorily, and the research, in its current form, is likely to draw little interest from the readers of this work.

There were a lot of grammatical errors in the manuscript. Many abbreviations used in the manuscript have been defined multiple times, which is not needed. Please re-read the manuscript and rectify those errors (some of them are highlighted in yellow). Please see additional comments in the attached PDF.
